# Enhancing site selection strategies in clinical trial recruitment using real-world data modeling

**Lars Hulstaert**[1]*, **Isabell Twick**[1], **Khaled Sarsour**[2], **Hans Verstraete**[3]

1 R&D Data Science & Digital Health, Janssen-Cilag GmbH, Neuss, North Rhine-Westphalia, Germany, 2 R&D Data Science & Digital Health, Janssen Pharmaceuticals, Titusville, New Jersey, United States of America, 3 R&D Data Science & Digital Health, Janssen Pharmaceutica NV, Beerse, Antwerp, Belgium

* lhulsta1@its.jnj.com

**Data Availability Statement:** The data underlying this article were provided by third parties (Komodo Health & IQVIA) under license and cannot be shared publicly. The source data for this study were licensed by Johnson & Johnson from

## Abstract

Slow patient enrollment or failing to enroll the required number of patients is a disruptor of clinical trial timelines. To meet the planned trial recruitment, site selection strategies are used during clinical trial planning to identify research sites that are most likely to recruit a sufficiently high number of subjects within trial timelines. We developed a machine learning approach that outperforms baseline methods to rank research sites based on their expected recruitment in future studies. Indication level historical recruitment and real-world data are used in the machine learning approach to predict patient enrollment at site level. We define covariates based on published recruitment hypotheses and examine the effect of these covariates in predicting patient enrollment. We compare model performance of a linear and a non-linear machine learning model with common industry baselines that are constructed from historical recruitment data. Performance of the methodology is evaluated and reported for two disease indications, inflammatory bowel disease and multiple myeloma, both of which are actively being pursued in clinical development. We validate recruitment hypotheses by reviewing the covariates relationship with patient recruitment. For both indications, the non-linear model significantly outperforms the baselines and the linear model on the test set. In this paper, we present a machine learning approach to site selection that incorporates site-level recruitment and real-world patient data. The model ranks research sites by predicting the number of recruited patients and our results suggest that the model can improve site ranking compared to common industry baselines.

## Introduction

### Background

Slow patient enrollment or failing to enroll the required number of patients is a disruptor of clinical trial timelines, leading to potential delays in drug approval, underpowered studies, the need to include additional study sites or even trial terminations [1–3]. Site selection, the process in which research sites, healthcare organizations and their associated investigators, are chosen to participate in clinical trials, is critical to enable timely recruitment.

Komodo Health and IQVIA, and hence we are not allowed to share the licensed data publicly. However, the same data used in this study are available for purchase by contracting with the database owners, Komodo Health (contact at: https://www.komodohealth.com/) and IQVIA (contact at: https://dqs.drugdev.com/help/contactUs). The authors did not have any special access privileges that other parties who license the data and contract with Komodo Health and IQVIA would not have. Further, in our efforts to enable use and reproducibility of the prediction model, we have provided detailed supporting material on covariates, and model hyperparameter settings.

**Funding:** The author(s) received no specific funding for this work.

**Competing interests:** All authors are employees of Janssen Research and Development, a unit of Johnson and Johnson family of companies. The work on this study was part of their employment. All authors hold pension rights from the company and own stock options. This does not alter our adherence to PLOS ONE policies on sharing data and materials.

Common site selection strategies use past trial data to assess how well a site would perform in a prospective clinical trial and different standardized and objective methods have been developed across industry and academia [4–8]. These methods include analyzing factors such as prior trial participation and performance, which are interrogated through database searches in investigator, site, and enrollment data sources. In certain cases, this process is complemented with epidemiologic and geographical analyses to create short lists of research sites that have both relevant research experience and direct access to a sufficiently large target patient population [5,9,10]. Research sponsors subsequently use site-level feasibility questionnaires to get estimates of expected recruitment for shortlisted sites–often resulting in an overestimation of their ability to recruit patients in trials [11]. It is hypothesized that this overestimation is because investigators do not have full protocol information and limited time for a thorough trial feasibility assessment [6].

Estimating site-level trial performance is a complex problem, further complicated by an increasingly competitive trial landscape and complex clinical trial designs [12]. In Table 1, different factors are summarized that have been reported in literature to influence site recruitment performance. A common challenge in site selection strategies is aggregating the information collected to make final decisions. The range of variables that impact the performance of a site require an effective measurement of the trade-offs that influence prospective recruitment performance [3].

Quantitative research in this field is limited by the volume of clinical trial data needed to generate meaningful recruitment insights. Typically, the impact of the reported site level factors on recruitment performance either is not validated, validated only on a small sample of studies or only with feasibility questionnaire data of a single study. The low power observed in these analyses signals that a cross-study analysis is needed to yield generalizable results [17].

Different statistical and machine-learning methods have been developed to estimate trial recruitment to address these challenges. Approaches differ in whether they predict enrollment at study [18,19] or study-site level [20,21], and whether enrollment is predicted at the start of the trial [18,20,21], or over the course of a trial's enrollment duration [20]. Several study related factors associated with trial enrollment have been studied (e.g. trial design, phase, sponsor, disease indication, competing trials recruiting similar patients, investigator experience and characterization) [20,21]. Existing study-site models examine variables that describe a small number of site-specific factors such as research experience, but these modeling approaches and their data are not tailored to the study indication and population. The use of

**Table 1. An overview of variables published in literature that are hypothesized to drive site recruitment performance.**

| Variable | References |
| --- | --- |
| Past site & investigator trial performance. | [3,6–8,13–16] |
| Past site & investigator trial experience. | [3,13–15,17] |
| Study population, procedures and treatments performed at hospital. | [2,3,6,7,14,16] |
| Time required to enroll the first subject in past studies. | [6,7,13,14] |
| Duration of recruitment period. | [17] |
| Site research capabilities and infrastructure. | [14–16] |
| Number of specialists and dedicated research staff. | [2,6,7,14,15] |
| Time available for research. | [6,7,14] |
| Interest in research and publication track record. | [3,15] |
| Language proficiency. | [15] |

electronic health record and claims data to characterize the available patient population for example remains limited [1,8].

## Objective

The goal of this work is to predict the number of patients enrolled at a clinical trial site, before the start of a trial's enrollment phase, using a broad set of indication-specific and site level characteristics. We explore a machine learning method that considers research experience, historical performance, patient availability and other site and investigator factors to make site-level enrollment predictions. The model predictions can be used in the operational planning phase prior to the start of a study when potential study sites are selected.

The methodology is validated in inflammatory bowel disease (IBD) and multiple myeloma (MM). Given the limited availability of real-world data sources with large-scale hospital coverage outside of the US, the analysis is limited to predict patient enrollment for US research sites. The approach aims to address the following research questions:

- How do approaches that leverage a broad range of (study)-site variables compare to baseline site selection strategies?

- Does the use of non-linear models boost the generalization performance vs. a simple linear Poisson regression model? Does the model benefit from capturing non-linear relations between covariates and the model target?

To allow for comparison with previously published model results, the models are also compared in their ability to identify the bottom and top 30% performing investigators [21].

## Materials and methods

### Data sources

The data used in this work is sourced from different systems that contain structured data related to studies, research sites, investigators, and patient populations. The following section provides a description of the data that is collected from these data sources. A summary of the data sources is provided in Table 2, describing the data type, provider, coverage & time frame.

Enrollment data from the DrugDev DataQuerySystem (DQS) is used to compute study-site level recruitment variables. DQS is a data platform that allows trial sponsors to share information on clinical trial recruitment and is used to capture study performance variables at site level such as the site open date, first and last subject enrolled date, the enrollment duration, and the number of patients who enrolled in a trial. The data is available for pharmaceutical trials across different disease indications. New data is made available monthly through DQS, and sponsors publish enrollment data from their systems consolidated onto a common data format once the study has been finalized.

For each site selection exercise, enrollment data is collected for so-called benchmark studies within a given indication from DQS. Benchmark studies are defined by manually reviewing the available studies within a given indication. These are further refined with to ensure benchmark trials are like prospective studies in terms of study phase, target indication, eligibility criteria, study duration and type of intervention.

In Fig 1, the benchmark studies that have been used across the two exercises are visualized across study phase and study indication. The enrollment data, in terms of number of patients enrolled and enrollment months, is shown in Fig 2. The site distribution across US states is shown in Fig 3, as well as the site open year distribution. The complete list of benchmark studies for each exercise is provided in S1 File.

**Table 2. An overview of the different data sources with a description of the data, coverage, and time frame.**

| Data type & provider | Description | Coverage & time frame |
|---|---|---|
| Real-world data; Komodo Health | Claims database containing open and closed claims; covering inpatient, outpatient, emergency department and institutional encounters. | High coverage of the US population with healthcare insurance coverage (Commercial, Managed Medicaid, Medicaid, Medicare Advantage, Dual Eligible). (300M+ patients—2016-present). |
| Enrollment data; DrugDev DataQuerySystem (DQS) | Trial recruitment database, containing site level patient recruitment data across clinical studies. | DQS covers pharmaceutical, sponsor-led trials–provided voluntarily as part of DrugDev Consortium (178k+ studies—1990-present). |
| Public study data sourced from ClinicalTrials.gov and aggregated by Komodo Health | Trial database prepared by Komodo Health by linking clinical trials from ClinicalTrials.gov data to healthcare providers using NLP-based provider matching. | Site & investigator level study participation data for privately and publicly funded clinical studies conducted in the US. (440k+ studies—initiated in 2000.). |
| Research publication data sourced from PubMed and aggregated by Komodo Health | Publication database prepared by Komodo Health by linking publications on Pubmed to healthcare providers using NLP- based provider matching. | Information on published manuscripts at investigator level on biomedical literature from MEDLINE, life science journals, and online books referenced in the National Library of Medicine. (35M+ citations—initiated in 1996.). |

Data from the Komodo Healthcare map, a real-world data (RWD) source with significant geographical coverage in the US, is used to characterize the study population. This data contains longitudinal patient claims data with information on prescribed drugs, diagnoses, procedures, and treatments.

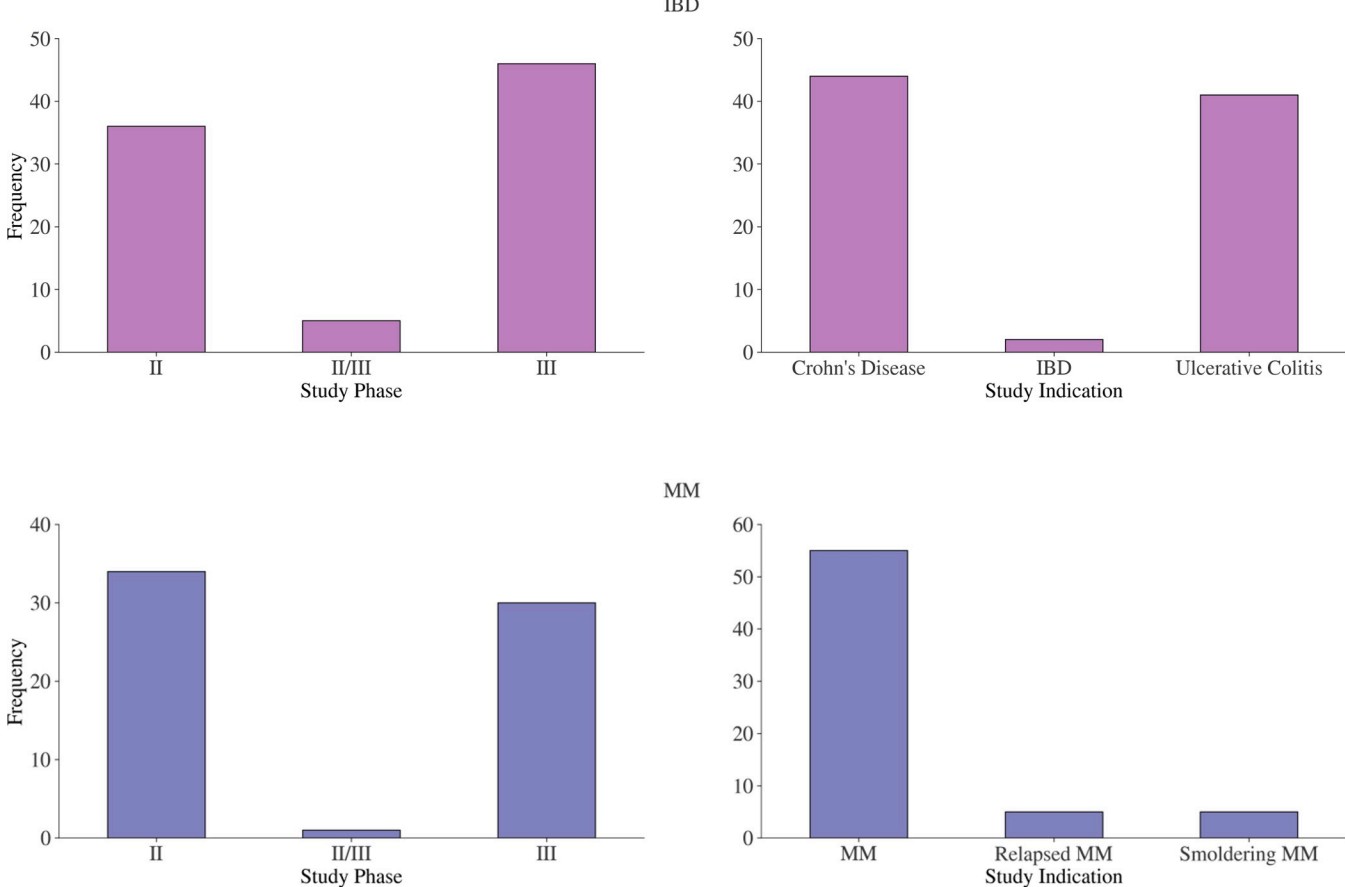

**Fig 1. Overview of number of benchmark studies across phase and indication.** An overview of the number of benchmark studies across study phase and study indication for resp. IBD and MM.

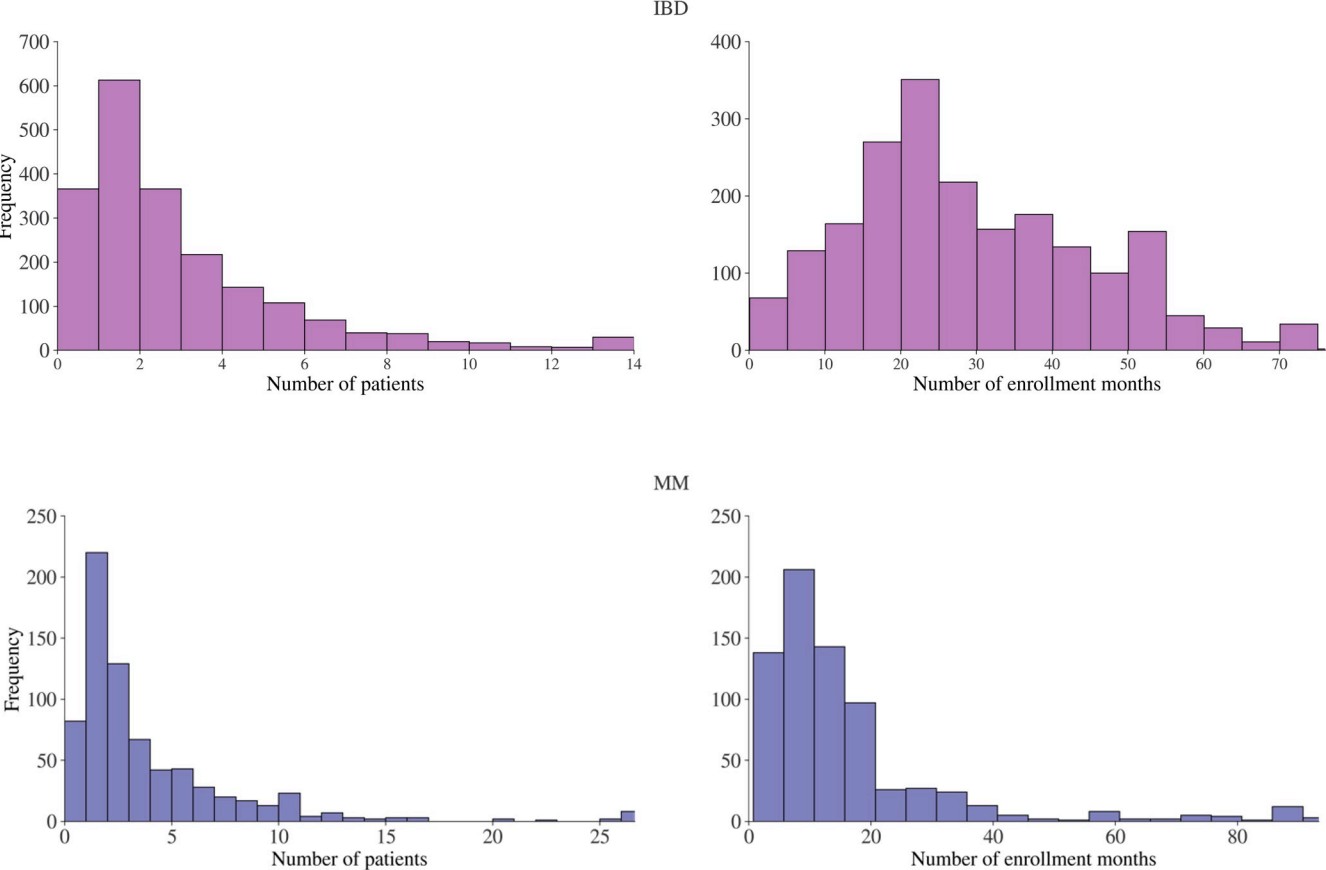

**Fig 2. Overview of number of patients enrolled and number of enrollment months across study-site combinations.** An overview of the number of patients enrolled and number of enrollment months per site across the benchmark studies for resp. IBD and MM.

Linkage of patients across longitudinal data (e.g., instances where a patient is treated at multiple institutions during a follow-up period and across healthcare providers) is performed by Komodo Health prior to sharing the data. Based on these complete patient journeys we can characterize the referral patterns across healthcare providers. Additional tables are provided by Komodo that describe provider level publications and trial participation. These tables have been processed in such a way that they can easily be linked to healthcare providers based on the National Provider Identifier (NPI) system [22].

Patient cohorts are created from common trial eligibility criteria from the benchmark studies to mimic the target population of the prospective and benchmark studies. Exact replication of the target patient population is often not possible with the available claims data. Patient outcomes and lab measurement results for example are typically not available in claims data, while they are often part of a trial's eligibility criteria. Patient cohort definitions are defined to mimic the general target patient population across benchmark studies. Patient populations are specified through an observation period, specialist type, patient age, diagnosis, drugs, and procedure codes. Inclusion criteria of benchmark studies are used to define a superset of relevant diagnosis, drugs, and procedures codes. These codes define a patient cohort that represents the broad patient population that is eligible for the benchmark studies. The cohort definitions for each exercise are shared in S2 File. The publication and trials databases are filtered only at indication level to capture the breadth of the research experience and interest of the HCO.

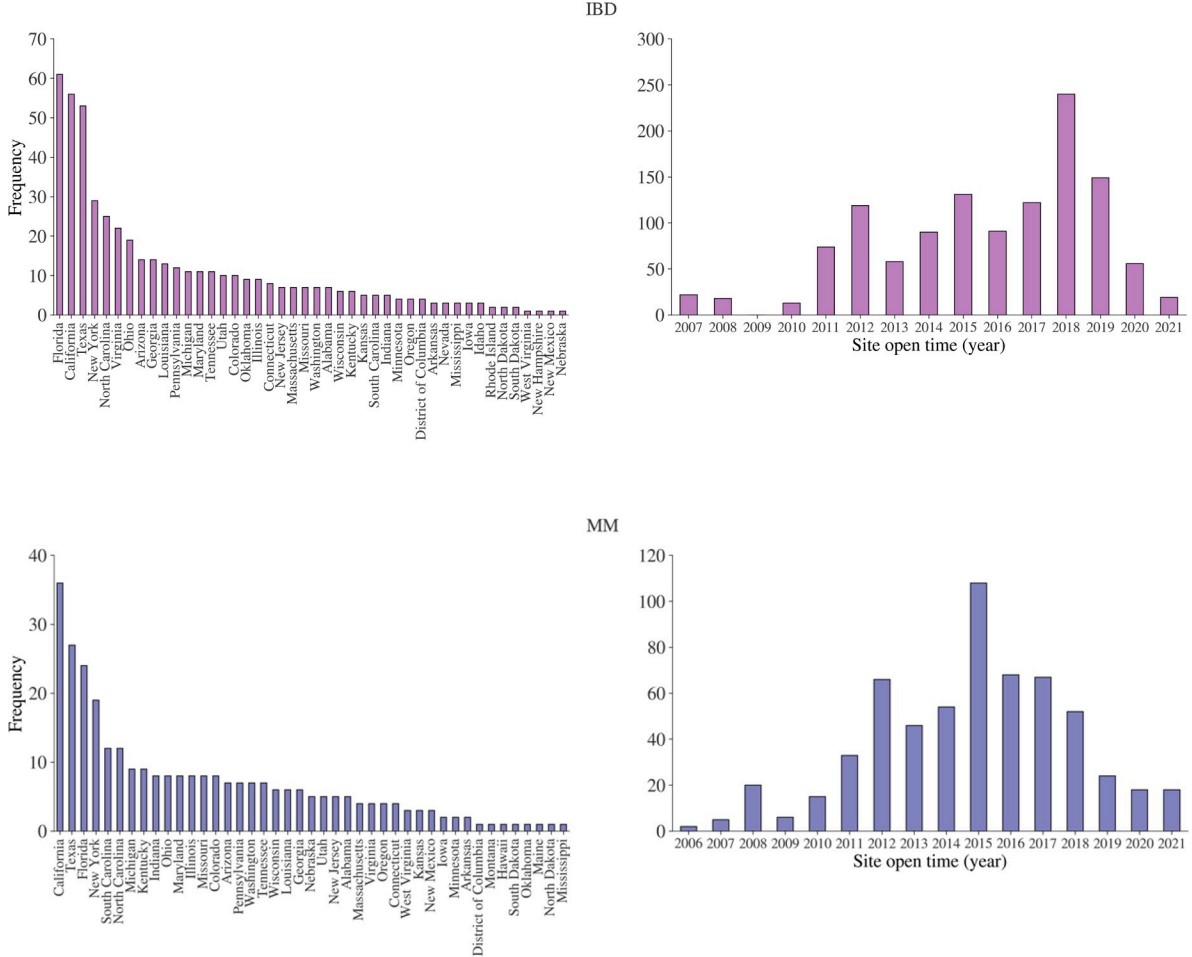

**Fig 3. Overview of number of sites across US states and open year.** An overview of number of sites across US states and site open year for resp. IBD and MM.

Real-world data is available at patient level but aggregated at healthcare provider (HCO) level. Different covariates are extracted at HCO level to characterize the available study population, procedures, treatments, staff, publications, and trial experience. The RWD and recruitment data sources are linked at healthcare organization (HCO) level. As DQS and Komodo use different HCO identifiers, manual validation is performed to ensure that each HCO is correctly linked across the data sources.

The real-world data that has been accessed for this study were deidentified in accordance with the Health Insurance Portability and Accessibility Act, and no personal health information was extracted. Therefore, no informed consent or institutional review board approval is required for this study.

## Outcome of interest and covariates

The outcome of interest, the enrollment at study-site level, is defined as the total number of recruited patients at a given site for a given study. A summary of the enrollment characteristics of the two exercises is provided in S1 Table. Covariates are constructed from enrollment and real-world data to characterize a site within the context of a study. From the enrollment data, historical performance and experience variables are generated, such as covariates that

summarize historical recruitment over a time window, the enrollment period, and the year when the site opened for a given study. The real-world data is used to build a set of indication-specific covariates that characterize study population, treatment, staff (physicians and specialists), referrals, publication, and trial participation.

Table 3 shows the different covariates that are generated and how they are constructed from the respective data source. Two types of covariates exist, those that characterize the site (site level covariate), are assumed to remain static over time and are not different across benchmark studies, and those that change over time and are unique at the study-site level (study-site level covariate). The covariate creation process was largely hypothesis driven, as defined in Table 1. Covariates are clipped for outliers outside the [0%, 97.5%] interval to the nearest value. Right-side clipping is used as the different covariates are gamma distributed. Missing values are imputed by 0 when a value is missing.

Although real-world data represents a broad set of patients that are potentially eligible for trial participation at any given time, its covariates are not aligned at the study-site level. While temporal alignment of RWD & recruitment data is possible based on the claim date and enrollment period for a site in each study, the real-world data is available only from 2016 onwards, while the benchmark studies start as early as 2006. As such, the cohort observation period is used instead to characterize the real-world clinical practice of a site. The variability in yearly calculations of the site level RWD covariates across the available data is sufficiently small, allowing them to be approximated as constant when averaged across the cohort observation period. Before 2016 it is not possible to validate this hypothesis which has the potential to introduce data bias in RWD covariates for studies conducted before 2016.

## Proposed approach

To predict site performance based on enrollment and real-world data covariates, different machine learning models have been developed. The number of recruited patients at study-site level are discrete counts that follow a Poisson distribution. The machine learning problem is defined as a Poisson regression problem where the enrollment months represent the exposure period.

We use a random train (80%) and test (20%) data split at site level to avoid the potential of a data distribution bias and corresponding impact on model generalization capabilities. The use of study specific variables is limited to ensure generalizability across studies and limit data leakage. A similar approach is used to perform cross-validation, using 5-fold cross-validation groups. In line with the Poisson modeling objective, models are compared with different regression metrics; mean absolute error (MAE), root mean squared error (RMSE), Spearman correlation coefficient are evaluated on both train and test set. The coefficient of determination ($R2$) is provided as reference, as the models are optimized on their ability to rank sites using the Spearman correlation coefficient.

We also assess whether the models succeed in identifying the top & bottom 30% of research sites in terms of enrollment, to allow for comparison with the results provided in prior work [21].

The regression outputs are converted into a ranked list and sites are grouped into two classes, top 30% and bottom 70%; top 70% and bottom 30%, based on whether they are part of the target group, respectively top 30% and bottom 30%. This group assignment is done with the actual and predicted enrollment counts to create the actual and predicted labels. We use the area-under-the-curve (AUC) classification metric to compare the different models on this classification task.

As there are no guidelines for systematic evaluation of site selection methods, the performance of new methods is compared to the median historical enrollment as baseline method.

**Table 3. Overview of the covariates.**

| Covariate | Variable description & construction | Source & type |
|---|---|---|
| Patients per site (target) | The number of recruited patients at a given site for a given study. | DQS study-site level |
| Enrollment months | Number of months a given site enrolled patients in the study.<br>The number of months is defined as the time between the site open date and the date of the last subject enrolled in the study. | |
| Site open year | The year when the site opened in the study. | |
| Median/EWMA of patients per site per month in the last 5 year | The median or exponential weighted moving average (EWMA) of the number of patients per month at a site across the studies the site participated in over the last 5 years.<br>The time window is indexed on the site open date for a given study. | |
| Median/EWMA rank in enrolled patients in the last 5 year | The median or EWMA of the rank of the site with respect to number of patients across studies the site participated in over the last 5 years.<br>The time window is indexed on the site open date for a given study. | |
| Median/EWMA/Sum of number of enrolled patients in the last 5 year | The median, EMWA or sum of number of patients recruited across studies the site participated in over the last 5 years.<br>The time window is indexed on the site open date for a given study. | |
| Number of claims | Number of unique claims (as defined by the diagnosis codes) at the site across the observation period. | RWD site level |
| Number of patients | Number of unique patients with a relevant claim (as defined by the diagnosis codes) at the site across the observation period. | |
| Number of treated patients | Number of unique patients with a relevant claim (as defined by the diagnosis codes) that have received treatment (as defined by medication and procedure codes) at the site across the observation period. | |
| Number of visits | Number of unique visits with a relevant claim (as defined by the diagnosis codes) at the site across the observation period. | |
| Visits per patient | Number of unique visits per patients with a relevant claim (as defined by the diagnosis codes) at the site across the observation period. | |
| Number of physicians | Number of unique physicians associated with relevant claims (as defined by the diagnosis codes) at the site across the observation period. | |
| Number of treating physicians | Number of unique physicians associated with relevant claims (as defined by the diagnosis codes) that treated patients (as defined by medication and procedure codes) at the site across the observation period. | |
| Number of specialists | Number of unique specialists (as defined by a list of specialists of interest) at the site across the observation period. | |
| Patient flow difference | Difference between number of incoming and outgoing patient referrals (as defined by counting the number of outgoing and incoming patient referrals with a relevant claim as defined by the diagnosis codes) at the site across the observation period. | |
| Site flow difference | Difference between number of sites that refer patients to the site vs the number of sites that the site refers patients to. | |

(*Continued*)

**Table 3.** (Continued)

| Covariate | Variable description & construction | Source & type |
|---|---|---|
| Number of publications | Max number of publications in the indication of interest by grouping across the physicians at the site. | PubMed site level |
| Number of ongoing trials | Number of ongoing trials at the site. | ClinicalTrials. gov site level |
| Number of Janssen trials | Number of Janssen trials at the site. | |
| Trial experience | A binary variable indicating whether a site had past trial experience | |

With this baseline, referred to as the median baseline, the median of the enrollment in train set is used to predict the enrollment of sites in the test set.

To reflect a common industry practice of using historical performance, we add a baseline method based on site-level historical enrollment, referred to as the site level baseline. With this baseline, the median of the historical enrollment of a site is used, to predict the enrollment of the site in other studies. If no historical enrollment data is available for a site, we impute the historical enrollment with the median historical enrollment in the train set.

## Covariate selection and model training

For each exercise, a linear Poisson regression model and two non-linear machine learning models, a RandomForest and an XGBoost model (v.1.7.2), are trained and compared with the median and site baseline [23]. We considered other non-linear models but didn't observe a significant difference in performance. The open-source framework Tune (v.0.1.5) [24] is used to train and perform hyperparameter tuning on the non-linear models. The range of hyperparameters across which the models are optimized is shared in S4 File, as well as the optimal set of hyperparameters for each experiment. In the hyperparameter optimization framework, a new set of hyperparameters is randomly sampled in each experiment and evaluated using cross-validation. Across 128 experiments, the set of optimal hyperparameters is identified for a given dataset.

The Shapley Additive exPlanations (SHAP) [25] algorithm is used to estimate the importance of the covariates and to determine the partial dependency relationship between covariates and enrollment. Manual covariate selection is performed by assessing covariate importance with the model trained on all covariates using the training data. Covariates with a variable importance, as defined by the covariate mean SHAP value, that is below 0.005 are removed from the covariate set. For each model, the selected set of covariates is defined in S3 File, which is a subset of the full set of covariates described in Table 3. To assess whether accuracy differences across modeling approaches are statistically significant a dependent t-test for paired samples is conducted on the model absolute error.

## Results

The model performance results across the different indications are shared in Table 4 for the test dataset. Train model performance results have been added to S2 Table for completeness. The different performance metrics are computed between the target and predicted enrollment. We compare a simple median baseline site selection strategy, with a more advanced 'site level' baseline. Finally, we compare the use of non-linear models with more simple linear modeling methods.

Significance testing has been applied (paired Student's t-test) to assess the significance in performance, as measured by mean absolute error difference between the models. For each

**Table 4. Performance metrics are computed between the actual and predicted enrollment.**

| Indication | Model | Test R2 | Test Spearman correlation coefficient | Test RMSE | Test MAE | Test Top 30% AUC | Test Bottom 30% AUC |
|---|---|---|---|---|---|---|---|
| IBD | Median Baseline | - | - | 2.54 | 1.76 | 0.50 | 0.50 |
| | Site Baseline | - | 0.13 | 3.16 | 2.14 | 0.51 | 0.55 |
| | Linear Model | 0.14 | 0.44 | 2.33 | 1.70 | 0.77 | 0.77 |
| | Random Forest | 0.19 | 0.46 | 2.26 | 1.64 | 0.78 | 0.81 |
| | XGBoost | **0.22** | **0.46** | **2.22** | **1.60** | **0.78** | **0.84** |
| MM | Median Baseline | - | - | 3.58 | 2.31 | 0.50 | 0.50 |
| | Site Baseline | - | 0.16 | 4.07 | 2.71 | 0.57 | 0.73 |
| | Linear Model | 0.10 | 0.34 | 3.24 | 2.52 | 0.71 | 0.76 |
| | Random Forest | **0.15** | 0.35 | 3.20 | 2.40 | **0.75** | **0.84** |
| | XGBoost | 0.13 | **0.37** | **3.19** | **2.40** | 0.74 | 0.79 |

experiment, the non-linear XGBoost model had a mean absolute error that was significantly lower than the linear models (p-value < 0.001). As no significant performance difference was observed among the non-linear models, only the XGBoost models are studied further.

We use Shapley values [25] to estimate covariate importance in the model in Fig 4. We also assess the relationship the model has learned between study-site level enrollment and the covariates of interest. Partial dependence plots, computed from the Shapley values, based on the XGBoost models, are used to visualize the relationship of a model covariate with the target variable. We explore the relationships between all selected covariates and the model target in Figs 5 and 6.

## Discussion

The proposed modeling approach is versatile and applicable across indications when sufficient benchmark studies are available, and the study population can be defined in a RWD cohort. The non-linear model performance improves the ability to rank the sites by expected enrollment, visible both in the increase of Spearman correlation coefficient and AUC on the test set, compared to the linear model and baselines. The ability to generate an accurate site level ranking allows trial organizers to accurately identify and prioritize top performing sites.

Comparing our results to the results of earlier data-driven site selection methodologies [21] is not straightforward due to variability in terms of study context (single indication vs multiple indications), methodology (site vs investigator ranking) and the evaluation approach (random vs study split) that is used. Comparing the top and bottom 30% AUCs of our methods with the published results, we observe that the average AUC on the test set (0.79) is higher compared to prior results (0.75), while our approach maintains a high level of interpretability on the relationship of study recruitment with the different covariates. Although the R2 remains low-to-modest, models have been optimized in the ability to rank sites, as expressed through the Spearman correlation coefficient and AUC.

Across the two experiments, there are important differences in the key covariates, highlighting the fact that different factors play a role in recruitment, depending on the indication. For instance, in trials targeting newly diagnosed patients like in the case for IBD, the research site must wait for patients to become available. In such cases, the recruitment period is an important covariate. On the other hand, for indications where patients are already undergoing treatment, such as MM, covariates that characterize the research setting, including the number of specialists, publications and ongoing trials are key covariates.

Regardless of the indication, past research experience, past high research performance and a high number of patients are consistently strong positive indicators of recruitment potential,

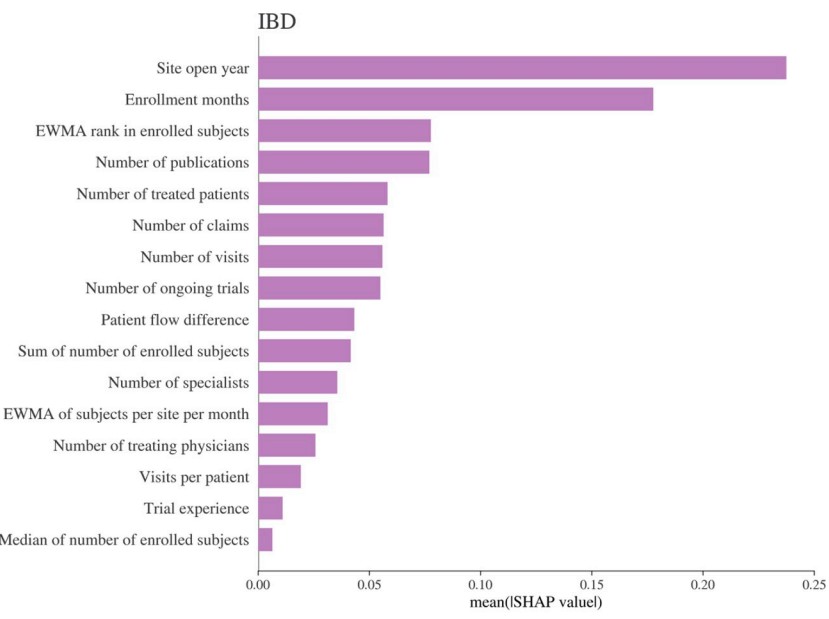

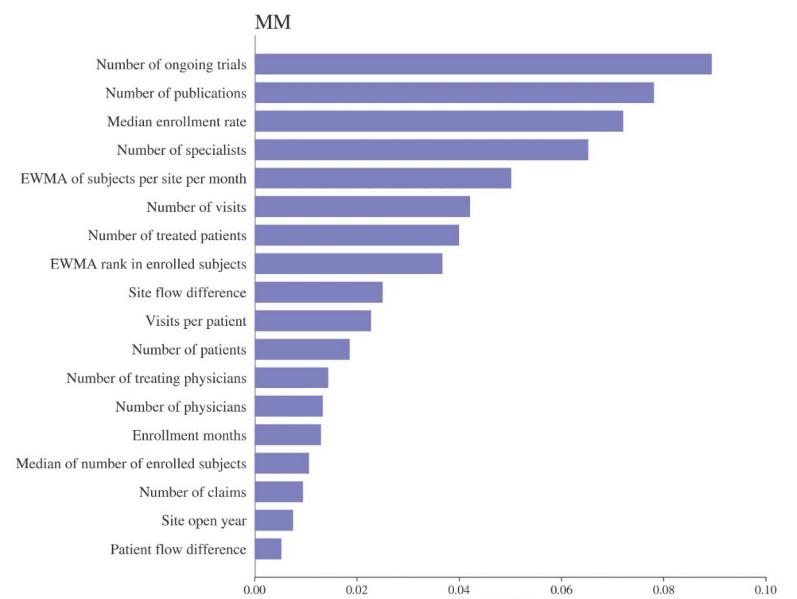

**Fig 4. Covariate importance of the selected covariates for the XGBoost models.** The mean SHAP value represents the average impact of a covariate on the model output magnitude.

aligned with previous research findings [1–17]. The site open year covariate captures the recruitment trend in a disease area over time and provides insights into the level of trial recruitment activity. In the case of IBD, the competition in the trial landscape has increased greatly, as highlighted by the high covariate importance. While the insights gained from these machine learning models are specific to each indication, they can serve to inform future trial designs and recruitment strategies.

The proposed site selection methodology represents a notable advancement; however, challenges with respect to data availability remain. The utility of real-world data for site selection

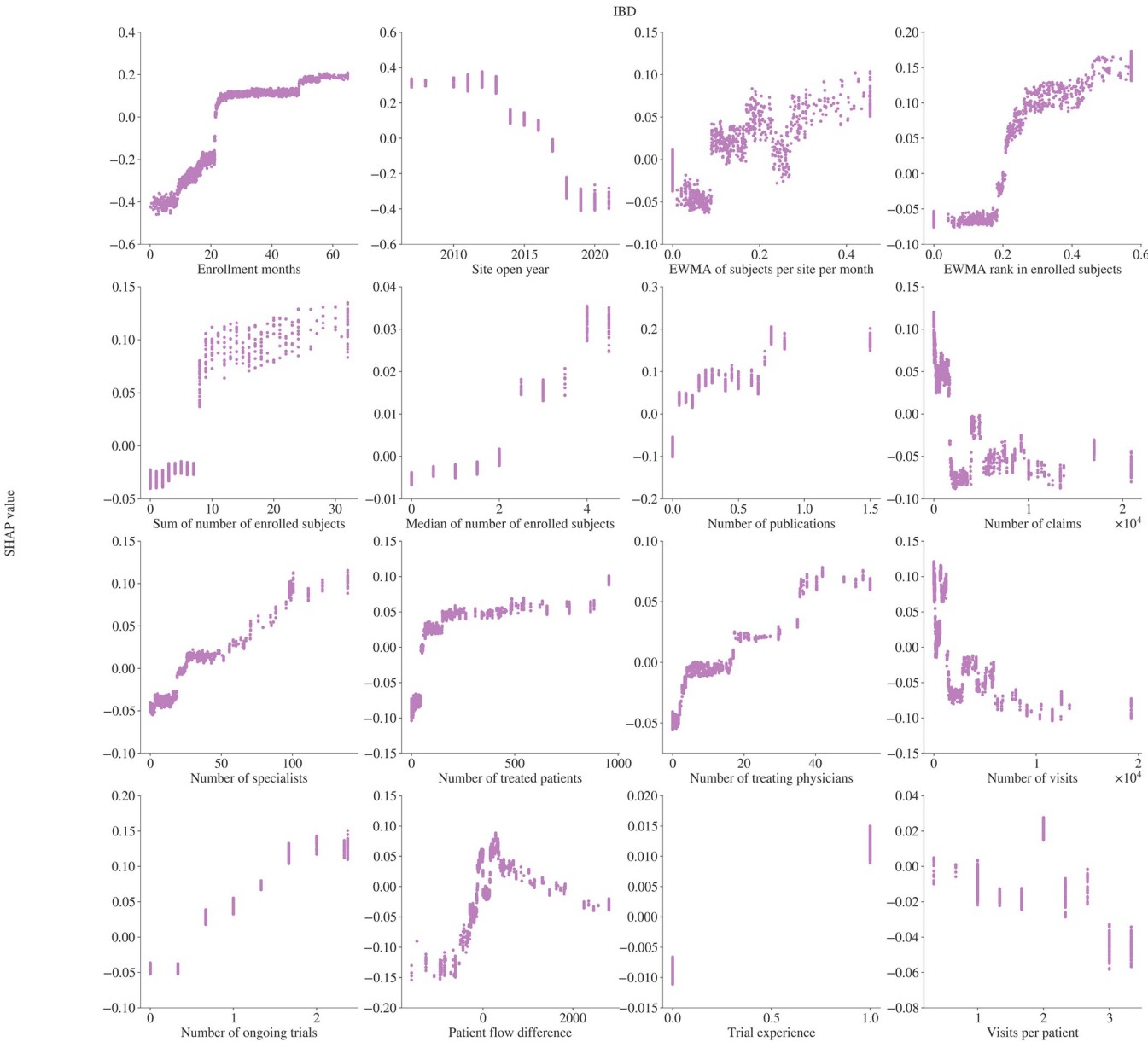

**Fig 5. Covariate dependence plots for the IBD XGBoost model.** The SHAP value represents the impact on the model output of a given covariate value.

relies on its availability across large geographical areas. At present, this approach is only viable in the United States. Moreover, due to the availability of US claims data from 2016 onwards, the data cannot be aligned to the study period of interest. Furthermore, the absence of large-scale linkage between claims data and electronic healthcare records, lab, and genomic data, poses challenges in the replication of study cohorts.

While expected recruitment is an important consideration in site selection strategies, it should not be the sole determinant in trial planning. Other factors, such as the overall experience collaborating with a research site and their research capabilities must also be considered. Additionally, sites with a diverse patient population need to be considered to improve the representativeness of the study population of clinical trials, and consequently the validity and

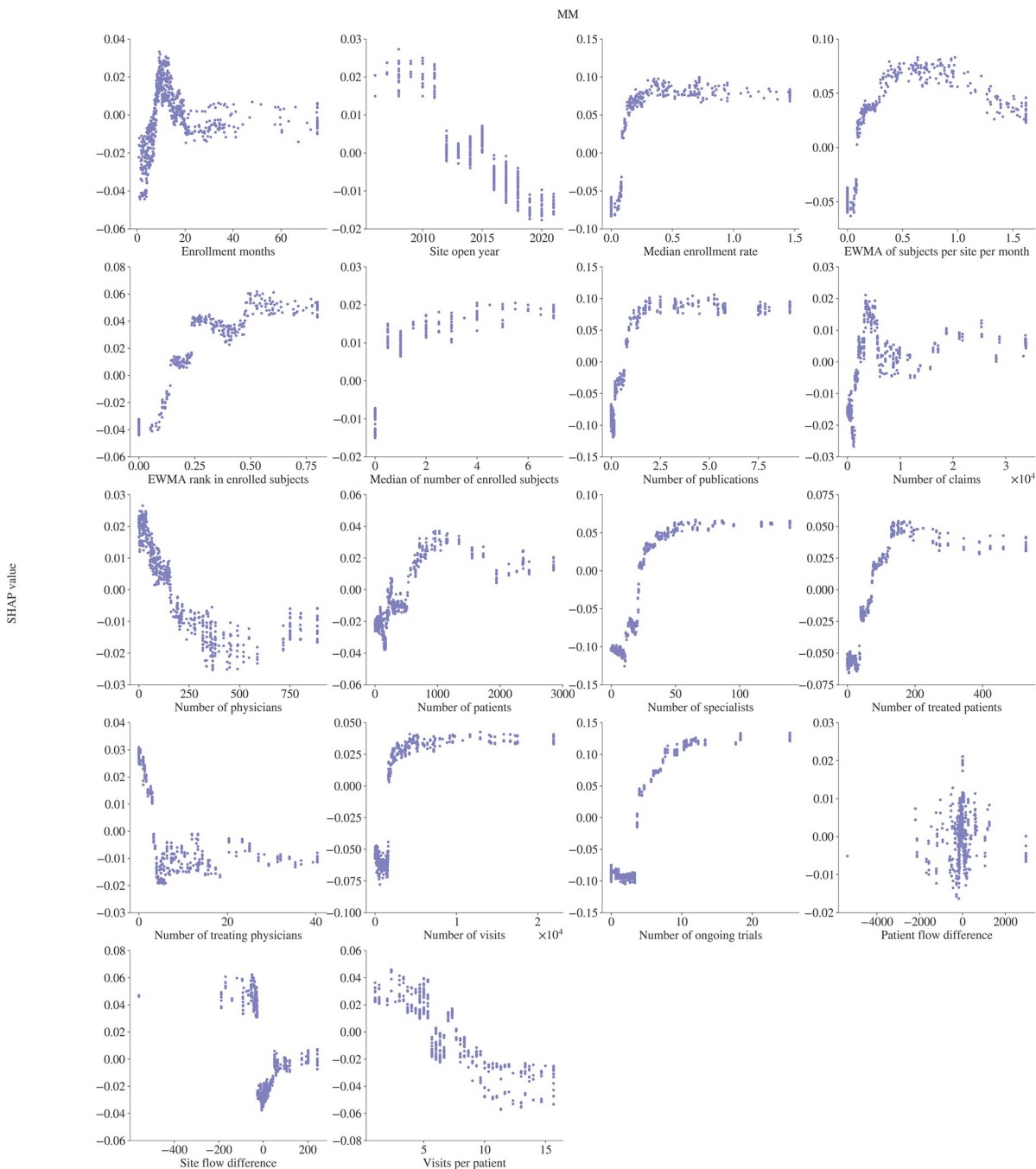

**Fig 6. Covariate dependence plots for the MM XGBoost model.** The SHAP value represents the impact on the model output of a given covariate value.

generalizability of clinical trials results. Nonetheless, within the United States, several barriers to diversity in clinical trial participation still exist [26]. Therefore, new, and diverse research sites, in addition to historically strong performing ones, need to be considered during site selection to ensure novel therapies are more broadly accessible geographically and across underrepresented populations.

## Conclusion

This work demonstrated empirically the importance of real-world data in predicting the patient recruitment of research sites in clinical trials. To the best of our knowledge, this is the first study that leverages machine learning methods and indication-level real-world data for site level enrollment prediction. This study adds to an improved understanding and quantitative validation of the factors that are critical to predict site study recruitment and a data-driven decision support system to help select and assess research sites for a proposed trial.

## Supporting information

**S1 File. List of benchmark studies per indication.**
(DOCX)

**S2 File. Real-world data cohort definition per indication.**
(DOCX)

**S3 File. Selected set of covariates per indication.**
(DOCX)

**S4 File. XGBoost hyperparameter grid and final model hyperparameters per indication.**
(DOCX)

**S1 Table. Summary of the enrollment statistics across the two experiments.**
(DOCX)

**S2 Table. Model performance on train set across the two experiments.**
(DOCX)

## Author Contributions

**Conceptualization:** Lars Hulstaert, Isabell Twick, Hans Verstraete.

**Data curation:** Lars Hulstaert.

**Formal analysis:** Lars Hulstaert.

**Funding acquisition:** Khaled Sarsour, Hans Verstraete.

**Investigation:** Lars Hulstaert, Hans Verstraete.

**Methodology:** Lars Hulstaert, Isabell Twick, Hans Verstraete.

**Supervision:** Khaled Sarsour, Hans Verstraete.

**Validation:** Lars Hulstaert.

**Visualization:** Lars Hulstaert.

**Writing – original draft:** Lars Hulstaert.

**Writing – review & editing:** Lars Hulstaert, Isabell Twick, Khaled Sarsour, Hans Verstraete.

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
