## [Decision Letter · Decision Letter 0]

18 Jan 2024

PONE-D-23-38747Enhancing site selection strategies in clinical trial recruitment using real-world data modelingPLOS ONE

Dear Dr. Hulstaert,

Thank you for submitting your manuscript to PLOS ONE. After careful consideration, we feel that it has merit but does not fully meet PLOS ONE’s publication criteria as it currently stands. Therefore, we invite you to submit a revised version of the manuscript that addresses the points raised during the review process.

We look forward to receiving your revised manuscript.

Kind regards,

Krit Pongpirul, MD, MPH, PhD.

Academic Editor

PLOS ONE

Journal Requirements:

2. Thank you for stating the following in the Competing Interests/Financial Disclosure* (delete as necessary) section: 

All authors are employees of Janssen Research and Development, a unit of Johnson and Johnson family of companies. The work on this study was part of their employment. All authors hold pension rights from the company and own stock options. This does not alter our adherence to PLOS ONE policies on sharing data and materials.

We note that one or more of the authors are employed by a commercial company: Janssen Research and Development, a unit of Johnson and Johnson family of companies.

“The funder provided support in the form of salaries for authors, but did not have any additional role in the study design, data collection and analysis, decision to publish, or preparation of the manuscript. The specific roles of these authors are articulated in the ‘author contributions’ section.”

Additional Editor Comments:

Please address the comments raised by both reviewers.

Reviewers' comments:

Reviewer's Responses to Questions

**Comments to the Author**

1. Is the manuscript technically sound, and do the data support the conclusions?

Reviewer #1: Yes

Reviewer #2: Yes

2. Has the statistical analysis been performed appropriately and rigorously? 

Reviewer #1: Yes

Reviewer #2: Yes

3. Have the authors made all data underlying the findings in their manuscript fully available?

Reviewer #1: No

Reviewer #2: Yes

4. Is the manuscript presented in an intelligible fashion and written in standard English?

Reviewer #1: Yes

Reviewer #2: Yes

5. Review Comments to the Author

Reviewer #1: -- Summary --

The authors present a ML-based model for ranking clinical trial sites by their expected recruitment potential for clinical trial site selection. They employ historical recruitment data combined with real-world claims data and additional public data sources for model training and evaluation. Models are trained, evaluated and modeling approaches compared for two diseases indications.

-- Comments --

- Definition of patient cohorts for benchmark studies: Suppl. file S2 indicates that multiple clinical studies were used as reference for the identified patient cohorts. Please comment on how the cohort inclusion criteria for the presented 'experiments' were derived from the inclusion criteria of these clinical studies. Presumably, inclusion criteria were not identical for all clinical studies, how was consensus reached for the experiments?

- Mismatch between RWD cohort observation period and recruitment data: "As such, the cohort observation period is used instead to characterize the real-world clinical practice of a site, and it is assumed to remain constant over time". Please discuss the potential biases that may be introduced by estimating site level covariates for the recruitment into studies performed much earlier (about 15-20 years, 1990 onwards) from rather recent RWD (2016 onwards).

- Manual covariate selection: What thresholds have been applied and which covariates were selected/removed? Does fig 4 show only the selected covariates?

- Authors state in the discussion that 'The non-linear model performance significantly improves the ability to rank the sites by expected enrollment ...', however, the methods section indicates that statistical significance testing has been performed on the MAE which is not a direct measure of ranking performance. Please explain.

- Fig 1-3: Please indicate the unit of 'frequency' in figs 1-3. 'Studies per year' in fig 1? But 'number of studies' in fig 2 and fig 3?

- Table 4: The reported results are labeled as 'Test'. Presumably this refers to the 20% of the 80/20 split mentioned in line 217. Please clarify the nomenclature. It would be interesting to see how robust these performance estimates are for different sample splits.

- Table S6 does not contain train model performances as mentioned in line 265.

Reviewer #2: This is a comprehensive machine learning study for site selection using trial data and real world data. A linear Poisson regression model and a non-linear XGBoost model were trained and compared with the baseline method. It clearly shows that XGBoost outperforms other methods. Shapley values were used to estimate covariate importance in the model. It helps interpret the relationship of study recruitment with the different covariates. Interestingly it highlights that different factors play a role in recruitment for different indications such as IBD and MM. Together these methods demonstrate the value of machine learning models in improving site selections.

Since there are many non-linear models besides XGBoost, it would be interesting to add a few more models to test how different algorithms perform in the two indications. Since the training/test data are ready, this seems straightforward. But if there are practical reasons why other algorithms were not selected, you may simply add the explanation.

6. PLOS authors have the option to publish the peer review history of their article (what does this mean?). If published, this will include your full peer review and any attached files.

Reviewer #1: No

Reviewer #2: **Yes: **Xiong Liu

---

## [Author Response · Author response to Decision Letter 0]

29 Jan 2024

Addressed reviewer comments in 'Response to Reviewers' letter. Please let us know if additional changes or revisions are required. 

Reviewer 1 Comments

Definition of patient cohorts for benchmark studies: Suppl. file S2 indicates that multiple clinical studies were used as reference for the identified patient cohorts. Please comment on how the cohort inclusion criteria for the presented 'experiments' were derived from the inclusion criteria of these clinical studies. Presumably, inclusion criteria were not identical for all clinical studies, how was consensus reached for the experiments? 

Updated line 166:

Inclusion criteria of benchmark studies are used to define a superset of relevant diagnosis, drugs, and procedures codes. These codes define a patient cohort that represents the broad patient population that is eligible for the benchmark studies.

Mismatch between RWD cohort observation period and recruitment data: "As such, the cohort observation period is used instead to characterize the real-world clinical practice of a site, and it is assumed to remain constant over time". Please discuss the potential biases that may be introduced by estimating site level covariates for the recruitment into studies performed much earlier (about 15-20 years, 1990 onwards) from rather recent RWD (2016 onwards).

Updated line line 212:

The variability in yearly calculations of the site level RWD covariates across the available data is sufficiently small, allowing them to be approximated as constant when averaged across the cohort observation period. Before 2016 it is not possible to validate this hypothesis which has the potential to introduce data bias in RWD covariates for studies conducted before 2016.

Manual covariate selection: What thresholds have been applied and which covariates were selected/removed? Does fig 4 show only the selected covariates?

Updated line 269:

Covariates with a variable importance, as defined by the covariate mean SHAP value, that is below 0.005 are removed from the covariate set. For each model, the selected set of covariates is defined in S4 File, which is a subset of the full set of covariates described in Table 3. 

Fig 4 description is adapted to “Covariate importance of the selected covariates for the XGBoost models” 

Authors state in the discussion that 'The non-linear model performance significantly improves the ability to rank the sites by expected enrollment ...', however, the methods section indicates that statistical significance testing has been performed on the MAE which is not a direct measure of ranking performance. Please explain. 

Updated line 306 and 320 to reflect the type of statistical testing that was applied.

Fig 1-3: Please indicate the unit of 'frequency' in figs 1-3. 'Studies per year' in fig 1? But 'number of studies' in fig 2 and fig 3?  

Descriptions of Fig 1-3 are adapted (line 138 to 145):

Fig 1. Overview of number of benchmark studies across phase and indication. An overview of the number of benchmark studies across study phase and study indication for resp. IBD and MM.

Fig 2. Overview of number of patients enrolled and number of enrollment months across study-site combinations. An overview of the number of patients enrolled and number of enrollment months per site across the benchmark studies for resp. IBD and MM.

Fig 3. Overview of number of sites across US states and open year. An overview of number of sites across US states and site open year for resp. IBD and MM.

Table 4: The reported results are labeled as 'Test'. Presumably this refers to the 20% of the 80/20 split mentioned in line 217. Please clarify the nomenclature. It would be interesting to see how robust these performance estimates are for different sample splits.

Updated line 224:

We use a random train (80%) and test (20%) data split at site level to avoid the potential of a data distribution bias and corresponding impact on model generalization capabilities.

Table S6 does not contain train model performances as mentioned in line 265.

Updated Table S6 to contain the train model performance.

Reviewer 2 Comments

Since there are many non-linear models besides XGBoost, it would be interesting to add a few more models to test how different algorithms perform in the two indications. Since the training/test data are ready, this seems straightforward. But if there are practical reasons why other algorithms were not selected, you may simply add the explanation. 

Updated Table 4 with results for RandomForest experiments.

Updated line 253:

For each exercise, a linear Poisson regression model and two non-linear machine learning models, a RandomForest and an XGBoost model (v.1.7.2), are trained and compared with the median and site baseline.

We considered other non-linear models but didn’t observe a significant difference in performance.

---

## [Decision Letter · Decision Letter 1]

23 Feb 2024

Enhancing site selection strategies in clinical trial recruitment using real-world data modeling

PONE-D-23-38747R1

Dear Dr. Hulstaert,

We’re pleased to inform you that your manuscript has been judged scientifically suitable for publication and will be formally accepted for publication once it meets all outstanding technical requirements.

Kind regards,

Krit Pongpirul, MD, MPH, PhD.

Academic Editor

PLOS ONE

Additional Editor Comments (optional):

Your responses to the comments from both reviewers are satisfactory. Please address the optional comments during the proof.

Reviewers' comments:

Reviewer's Responses to Questions

**Comments to the Author**

1. If the authors have adequately addressed your comments raised in a previous round of review and you feel that this manuscript is now acceptable for publication, you may indicate that here to bypass the “Comments to the Author” section, enter your conflict of interest statement in the “Confidential to Editor” section, and submit your "Accept" recommendation.

Reviewer #1: All comments have been addressed

Reviewer #2: All comments have been addressed

2. Is the manuscript technically sound, and do the data support the conclusions?

Reviewer #1: Yes

Reviewer #2: Yes

3. Has the statistical analysis been performed appropriately and rigorously? 

Reviewer #1: Yes

Reviewer #2: Yes

4. Have the authors made all data underlying the findings in their manuscript fully available?

Reviewer #1: No

Reviewer #2: Yes

5. Is the manuscript presented in an intelligible fashion and written in standard English?

Reviewer #1: Yes

Reviewer #2: Yes

6. Review Comments to the Author

Reviewer #1: Thank you for addressing the comments. Please consider the following questions and recommendations below optional:

- You mention in your answers (update line 212) that "The variability in yearly calculations of the site level RWD covariates across the available data is sufficiently small, allowing them to be approximated as constant when averaged across the cohort observation period. Before 2016 it is not possible to validate this hypothesis which has the potential to introduce data bias in RWD covariates for studies conducted before 2016."

-> Readers may be interested in how this hypothesis been verified for the period from 2016.

- Thank you for clarifying the meaning of 'frequency' in the captions of Figs 1-3.

-> Modifying the figures' y-axis accordingly would greatly facilitate the reading in my opinion.

Reviewer #2: The authors have adequately addressed my comments. The paper is now in a good shape for publication.

7. PLOS authors have the option to publish the peer review history of their article (what does this mean?). If published, this will include your full peer review and any attached files.

Reviewer #1: No

Reviewer #2: **Yes: **Xiong Liu

---

## [Editor Report · Acceptance letter]

29 Feb 2024

PONE-D-23-38747R1 

PLOS ONE

Dear Dr. Hulstaert, 

I'm pleased to inform you that your manuscript has been deemed suitable for publication in PLOS ONE. Congratulations! Your manuscript is now being handed over to our production team.

Kind regards, 

on behalf of

Assoc. Prof. Dr. Krit Pongpirul 

Academic Editor

PLOS ONE